# DIAMetAlyzer allows automated false-discovery rate-controlled analysis for data-independent acquisition in metabolomics

Oliver Alka [1,2✉], Premy Shanthamoorthy[3,4], Michael Witting [5,6,7], Karin Kleigrewe [8], Oliver Kohlbacher [1,2,9] & Hannes L. Röst [3,4,10✉]

The extraction of meaningful biological knowledge from high-throughput mass spectrometry data relies on limiting false discoveries to a manageable amount. For targeted approaches in metabolomics a main challenge is the detection of false positive metabolic features in the low signal-to-noise ranges of data-independent acquisition results and their filtering. Another factor is that the creation of assay libraries for data-independent acquisition analysis and the processing of extracted ion chromatograms have not been automated in metabolomics. Here we present a fully automated open-source workflow for high-throughput metabolomics that combines data-dependent and data-independent acquisition for library generation, analysis, and statistical validation, with rigorous control of the false-discovery rate while matching manual analysis regarding quantification accuracy. Using an experimentally specific data-dependent acquisition library based on reference substances allows for accurate identification of compounds and markers from data-independent acquisition data in low concentrations, facilitating biomarker quantification.

[1] Department of Computer Science, Applied Bioinformatics, University of Tübingen, Tübingen, Germany. [2] Institute for Bioinformatics and Medical Informatics, University of Tübingen, Tübingen, Germany. [3] Terrence Donnelly Centre for Cellular & Biomolecular Research, University of Toronto, Toronto, Canada. [4] Department of Molecular Genetics, University of Toronto, Toronto, Canada. [5] Metabolomics and Proteomics Core, Helmholtz Zentrum München, Neuherberg, Germany. [6] Research Unit Analytical BioGeoChemistry, Helmholtz Zentrum München, Neuherberg, Germany. [7] Chair of Analytical Food Chemistry, School of Life Sciences Weihenstephan, Technical University of Munich, Freising, Germany. [8] Bavarian Center for Biomolecular Mass Spectrometry, Technical University of Munich, Freising, Germany. [9] Institute for Translational Bioinformatics, University Hospital Tübingen, Tübingen, Germany. [10] Department of Computer Science, University of Toronto, Toronto, Canada. ✉email: oliver.alka@uni-tuebingen.de; hannes.rost@utoronto.ca

Mass spectrometry (MS) is a flexible tool that allows for the acquisition of data in either an untargeted or a targeted fashion. While the untargeted approach aims at detecting as many metabolites as possible, the targeted approach focuses on the most accurate quantification of a small subset of metabolites. Thus, targeted methods such as Multiple Reaction Monitoring (MRM) or Parallel Reaction Monitoring (PRM) are limited in analyte coverage but provide a precise quantification. Untargeted approaches often use data-dependent acquisition (DDA), which selects a large number of metabolites for fragmentation in a data-driven manner, based on the precursor selection. Data-independent acquisition (DIA) cycles through a series of predetermined mass ranges (DIA or SWATH windows) to acquire a high-resolution MS2 spectrum, thus boosting reproducibility by sampling the entire mass range. This allows for the systematic, unbiased acquisition of fragmentation spectra, at the cost of acquiring highly multiplexed spectra since the mass isolation range for each DIA window is generally larger than for other methods. A comparison of DDA and DIA data acquisition revealed that DDA excels in MS2 spectrum quality, whereas DIA shows a better performance in quantitative precision and MS2 spectrum coverage[1]. A major challenge of DIA for the field is the measurement of multiplexed spectra, which are considered lower quality.

Two distinct strategies exist to analyse DIA metabolomics data. Most of the current algorithms use an untargeted strategy based on deconvolution and either specialize in identification via spectral library search or in quantification via targeted extraction based on their deconvoluted pseudo-MS2 spectra[2–4].

In a targeted analysis strategy, the compounds to quantify are defined in advance. This requires knowledge of suitable analyte assays, i.e., retention times, and precursor masses with corresponding fragment masses (transitions). These transitions are collected in a so-called assay library which is then used to produce fragment-level extracted ion chromatograms (XICs) for each analyte fragment ion around the expected chromatographic retention time. These XICs (one for each fragment ion) then have to be verified for quality and compared with an internal (spiked-in) or external standard, which is currently a laborious and manual task that requires specialized expertize and training. While both the creation of assay libraries for DIA analysis and the processing of XICs has been automated in other fields[5], this is not the case in metabolomics. Additionally, a main challenge in targeted metabolomics is the detection of false positive metabolic features in the low signal-to-noise ranges of DIA results that are unable to be filtered[1].

Here, we present a workflow based on the targeted strategy which solves these issues, first integrating a complete end-to-end pipeline including assay library generation into a widely used software suite (OpenMS[6]) and secondly implementing a procedure to estimate robust and accurate false-discovery rates (FDRs) for DIA metabolomics. Our DIAMetAlyzer software combines DDA and DIA metabolomics data by deriving libraries based on high quality DDA MS2 spectra with few interferences and then subsequently uses DIA to perform quantification, exploiting the improved MS2 coverage and superior quantification performance of DIA[1]. Fully automated construction of the assay library permits the discovery and quantification of unknown metabolites and still achieves the quantification accuracy of a manually curated targeted approach. A combination of semi-supervised machine learning and on-the-fly decoy generation permits the estimation of statistically well-calibrated FDRs for the resulting data sets.

## Results

**DIAMetAlyzer workflow.** The workflow takes advantage of an experiment-specific assay library curated based on available DDA data and is thus tailored to a specific question and instrument. In metabolomics, annotating fragment ions with the underlying structure is not trivially possible, unlike proteomics. We use SIRIUS to annotate fragments using their compositional fragmentation tree approach[7,8]. The method models the fragmentation process based on available MS2 spectra and the chemical composition of the precursor[9]. In proteomics, decoys can then be generated using common approaches to alter the peptide sequence to determine the FDR[10]. Following the idea of a target-decoy approach, we use Passatutto as the basis for re-rooting of fragmentation trees to generate high-quality decoys[11]. The resulting target-decoy assay library allows for the targeted extraction and scoring of targeted transitions from the DIA data with FDR control.

The workflow follows multiple steps (Fig. 1). Candidate identification. Feature detection, adduct grouping, and accurate mass search are applied to DDA data. Library construction. The knowledge determined about the compound identification, potential adducts, and the corresponding fragment spectra are used to perform fragment annotation via compositional

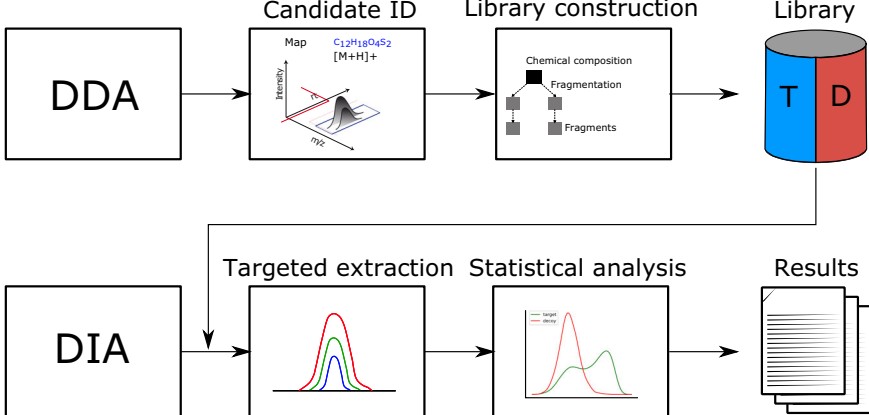

**Fig. 1 DIAMetAlyzer - a pipeline for assay library generation and targeted analysis with statistical validation.** DDA data is used for candidate identification containing feature detection, adduct grouping and accurate mass search. Library construction uses fragment annotation via compositional fragmentation trees (SIRIUS) and decoy generation using a fragmentation tree re-rooting method (Passatutto) to create a target-decoy assay library. This library is used in a second step to analyse metabolomics DIA data by performing targeted extraction (OpenSWATH), scoring and statistical validation (PyProphet).

fragmentation trees by using SIRIUS and extract transitions to build an assay library. FDR estimation is based on the target-decoy approach, with decoys being generated using the recently proposed re-rooting of fragmentation trees by Passatutto, which reduces bias in decoy generation. Targeted extraction. The assay library is then used to analyze DIA data. Targeted extraction involves chromatogram extraction and peak group scoring in OpenSWATH[5] (Supplementary Fig. 1). We modified Open-SWATH to support targeted extraction of metabolomics data. This feature is included in the latest release of OpenSWATH (see online documentation). Here, chromatograms in a user-specified retention time window are extracted from the DIA data based on the transition entries in the assay library. They encompass precursor and its isotope traces as well as the specified MS2 fragment traces. All extracted traces are grouped in so-called peak groups, which represent a possible analyte with MS1 and MS2 traces. For each peak group, a score matrix is generated based on different scores, such as co-elution and chromatogram shape. A detailed description of the OpenSWATH scores can be found in the original publication. Statistical validation. FDR estimation originated from the increasing amounts of data in the genomics field. It is the expected ratio of false positive classifications (false discoveries) to the total number of positive classifications. The "discovery" stands for the items that you label as "positive", and hence could be true positives or false positives, in the gene expression sense as the genes that you label as differentially expressed[12]. In 2007, Elias et al. introduced the concept of target-decoy FDR in proteomics[10], where it is used to distinguish correct from incorrect peptide identifications. In the targeted field experimental specific targets and decoys are added to the assay library (prior knowledge database) used for targeted extraction. For available targets and decoys, peak groups are extracted and scored. We use semi-supervised learning to build a composite score (discriminant score) out of individual peak group scores and estimate q-values by fitting a null distribution using a version of PyProphet adopted to metabolomics[13–15]. To prevent over-fitting, we chose a straightforward linear model (LDA) for target-decoy discrimination using peak group scores with a low cross-correlation, which resulted in an excellent performance on our benchmark dataset (Supplementary Fig. 2).

**FDR filtering and library coverage**. To assess FDR estimation accuracy and quantification performance, the developed pipeline was used for assay library generation and the subsequent analysis of the benchmark dataset (Methods). The analysis was performed automatically via DIAMetAlyzer and benchmarked against the manually annotated ground truth extracted via Skyline[16]. Using the DIAMetAlyzer workflow, we were able to reduce the number of false positive peak groups by 91% (from 1471 to 125) when applying a 5% FDR threshold to our results (Fig. 2a). The number of true positive peak groups were only reduced by 12% by the filtering step (from 3479 to 3071). Applying a 1% FDR filter, false positive peak groups were reduced by 98% (from 1471 to 19), and true positive peak groups by 28% (from 3479 to 2523). This demonstrates that our workflow can reduce the number of false positive detections/quantifications through an accurate target-decoy based false-discovery rate approach for DIA data analysis in metabolomics.

Following the pipeline from the start, first, an assay library was generated using reference mixes (Agilent Pesticide Mix, APM) diluted in solvent, measured using DDA acquisition and analyzed using the DIAMetAlyzer workflow (Fig. 2b). Since the goal was accurate identification and quantification, only high-quality assays were included in the library. In addition to 9% undetected pesticides, we filtered 14% of compounds that could not be detected

via MS1 or did not possess a valid MS2 spectrum (fewer than four peaks, to allow for fragment annotation). In the library construction step, filtering by the number of transitions greatly affects the coverage of metabolites (three transitions: 60% coverage, two transitions: 71% coverage, one transition: 77% coverage). By using data from multiple collision energy ranges (20–50 eV and 50–80 eV), coverage of the assay library can be increased by 11% to 71% (three transitions) (Supplementary Fig. 3).

To ensure the development of a high-quality assay library, theoretical simulations were used to determine the number of transitions required to reduce ambiguity and improve the number of unique identifications. Using the pesticides dataset with the NIST 17 LC/MS library as a combined background metabolome, MS methods were simulated with varying accuracy for both the precursor and fragment m/z windows, while alternating the selected number of transitions for each compound. Scoring both MS levels using three transitions increased the number of uniquely identified compounds in our simulation by ~2.8-fold and ~1.5-fold in comparison to MS1-only and MRM-based analyses, demonstrating the importance of both high-resolution MS1 and MS2 data (Supplementary Fig. 4a, rightmost bars). Based on these results, an assay library with three transitions has been chosen for downstream analyses.

Second, the developed assay library was used to perform the analysis of 30 DIA samples with APM spiked-in human blood plasma acquired in DIA mode using sequential window acquisition of all theoretical mass spectra (SWATH) (Supplementary Table 1). The pesticide mix was measured in triplicates and spiked into human plasma in a 4-fold dilution series, spanning over five orders of magnitude in dynamic range (Supplementary Table 2, Supplementary Fig. 5). Data was measured in a 10-step concentration series at two collision energy ranges. The targeted extraction was performed automatically via DIAMetAlyzer and benchmarked against the manually annotated ground truth extracted via Skyline.

**Accuracy of FDR estimation**. To evaluate the accuracy of our FDR estimates, the automatic and manual analyses were compared to determine the deviation of the ground truth FDR from the FDR estimated by the DIAMetAlyzer (Fig. 3a). We found that the FDR estimated by fragmentation tree re-rooting is slightly conservative, with a slight overestimation for data acquired at lower ranges of collision energy (20–50 eV). In comparison, FDR estimates for data acquired at higher ranges of collision energy (50–80 eV) demonstrated an increased abundance of overlapping fragments, resulting in more ambiguous analyses (Supplementary Fig. 6). To assess the accuracy of the classifier, we determined the precision and recall based on different estimated FDR thresholds using the best peak group rank (Fig. 3b). Our approach produced an area under the precision-recall curve (AUC) of 0.96, resulting in over 75% recall at 95% precision (or 5% FDR).

**Quantification performance**. To determine the quantification performance, the results were filtered using a 5% FDR threshold and normalized for each metabolite adduct combination by the intensity of their highest concentration. More than half of the initial metabolites could be detected at half maximal dilution (1:1,024), based on the last dilution step a metabolite was observed in (Fig. 3c, Supplementary Fig. 7). The limit of detection of the individual metabolites were assessed using the unfiltered results, based on an S/N threshold of 10 (Supplementary Table 3). Comparing the quantification of manual and automatic analyses, the precision of the automated method matches manual analysis and outperforms it in some dilution steps (Fig. 3d). In all technical replicates, the median coefficient of variation (CV) of non-

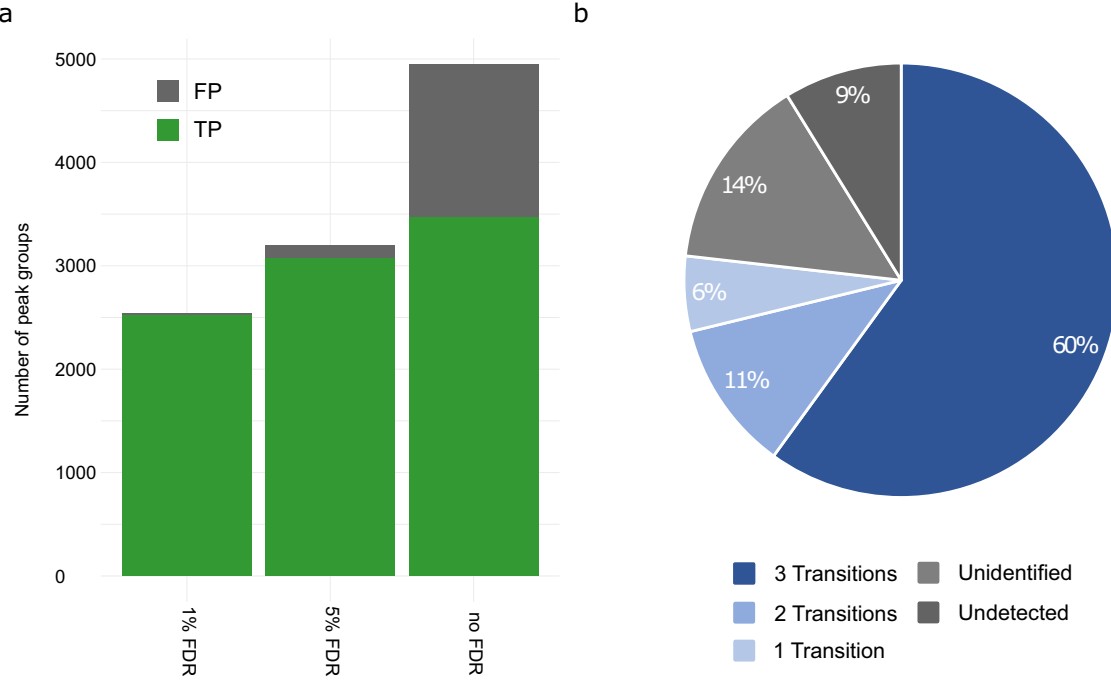

**Fig. 2 FDR filtering and library coverage. a** Peak groups detected and quantified by DIAMetAlyzer in a APM spiked-in human blood plasma dilution series (SWATH - 30 samples) filtered by different FDR thresholds. Without FDR filtering (no FDR), we detected and quantified the highest number of true positive peak groups (TP; $n = 3479$), but also the highest number of false positive peak groups (FP; $n = 1471$). At 5% FDR, 3071 true peak groups and 125 false positives were quantified (3.9%). At 1 % FDR the true positive peak groups were further reduced ($n = 2523$), so were the false positives ($n = 19$; 0.7%). **b** Individual pesticide mixes in solvent (around 30 pesticides each) were used to construct the target-decoy assay library. Stringent filtering allows high-quality assays to be used in library construction: Around 9% of the pesticides could not be detected in the data. An additional 14% were not identified via MS1 or did not possess a valid MS2 spectrum (4+ peaks, to allow for fragment annotation). 77% of the pesticides were automatically detected, identified, and annotated. In the library construction step, filtering by the number of transitions greatly affects the coverage of metabolites (three transitions: 60% coverage, two transitions: 71% coverage, one transition: 77% coverage).

normalized quantified signals was smaller than 0.2 (Fig. 3e, Supplementary Fig. 8).

**Comparison to state-of-the-art algorithms.** To benchmark the performance of the DIAMetAlyzer against state-of-the-art analysis algorithms, we compared it to MS-DIAL[2] and MetaboDIA[17]. MS-DIAL is a tool specialized in untargeted SWATH analysis based on spectral deconvolution, using computationally constructed pseudo-MS2 spectra for identification via spectral library search. Similar to the DIAMetAlyzer, the functionality of MS-DIAL is dependent on the spectral library space provided to the software. Here, to allow a fair comparison between the tools, we used our ground truth APM dataset specifying our assay library as a spectral library (Supplementary Fig. 9). The DIAMetAlyzer was able to identify 156 true positives and 3 false positive compounds in comparison to the ground truth (at 5% FDR). MS-DIAL was able to identify 84 true positives, 5 false positives and was not able to identify 70 compounds (false negatives). In this setting, we could show the advantage of the DIAMetAylzer targeted extraction strategy with false-discovery rate control based on reference compounds in comparison to untargeted deconvolution.

MetaboDIA is a tool capable of building a consensus MS/MS library based on DDA data using MS-based identification and subsequent quantification via DIA-MS/MS in a non-targeted manner. We used a publicly available age-related macular degeneration (AMD) data set (MetaboLights accession MTBLS417) along with HMDB[18] and LIPIDMAPS[19] for identification via accurate mass to construct a library with each tool. The library was filtered for features found in at least 20% of samples with a minimum of three MS/MS peaks available.

Libraries generated by both tools show a significant overlap (66%) of features with agreeing on molecular formula, adducts, and retention time (Fig. 4a). DIAMetAlyzer generates a larger number of features compared to MetaboDIA (46 % improvement; 695 compared to 476). Differences between the two libraries are likely attributable to improved feature detection and more stringent filtering in the assay library creation step in our pipeline.

Using our targeted quantification with the generated libraries, we were able to quantify almost twice as many features (811 vs. 440) with our library in comparison to the MetaboDIA library (Fig. 4b, Supplementary Fig. 10). When restricting quantification to identified features, DIAMetAlyzer could still increase quantification by 25% compared to a MetaboDIA-derived library. Interestingly, we found 144 features uniquely identified by MetaboDIA from the DDA data, allowing us to build a combined library which results in a total of 682 quantified features. These exclusive features were either not detected by our pipeline or were filtered out in the assay library generation step. Additional details on the feature detection, feature linking, and quantitative comparison with MetaboDIA are given in the supplementary material (Supplementary Figs. 11–14).

To assess the biological significance, we used the quantified features from the DIAMetAlyzer workflow at a 5% FDR ($FDR_{DIAMetAlyzer}$). LIMMA[20] was used with a Benjamini & Hochberg correction for multiple testing[12] to identify differentially expressed features between the conditions control, choroidal neovascularization (CNV), and polypoidal choroidal neovascularization (PCV). We found a total of 118 differentially expressed features ($FDR_{limma} < 0.05$), comparable to the 113 features found using our workflow together with the MetaboDIA library. We were able to report additional differentially expressed features using the combined library (162 features) and found the largest

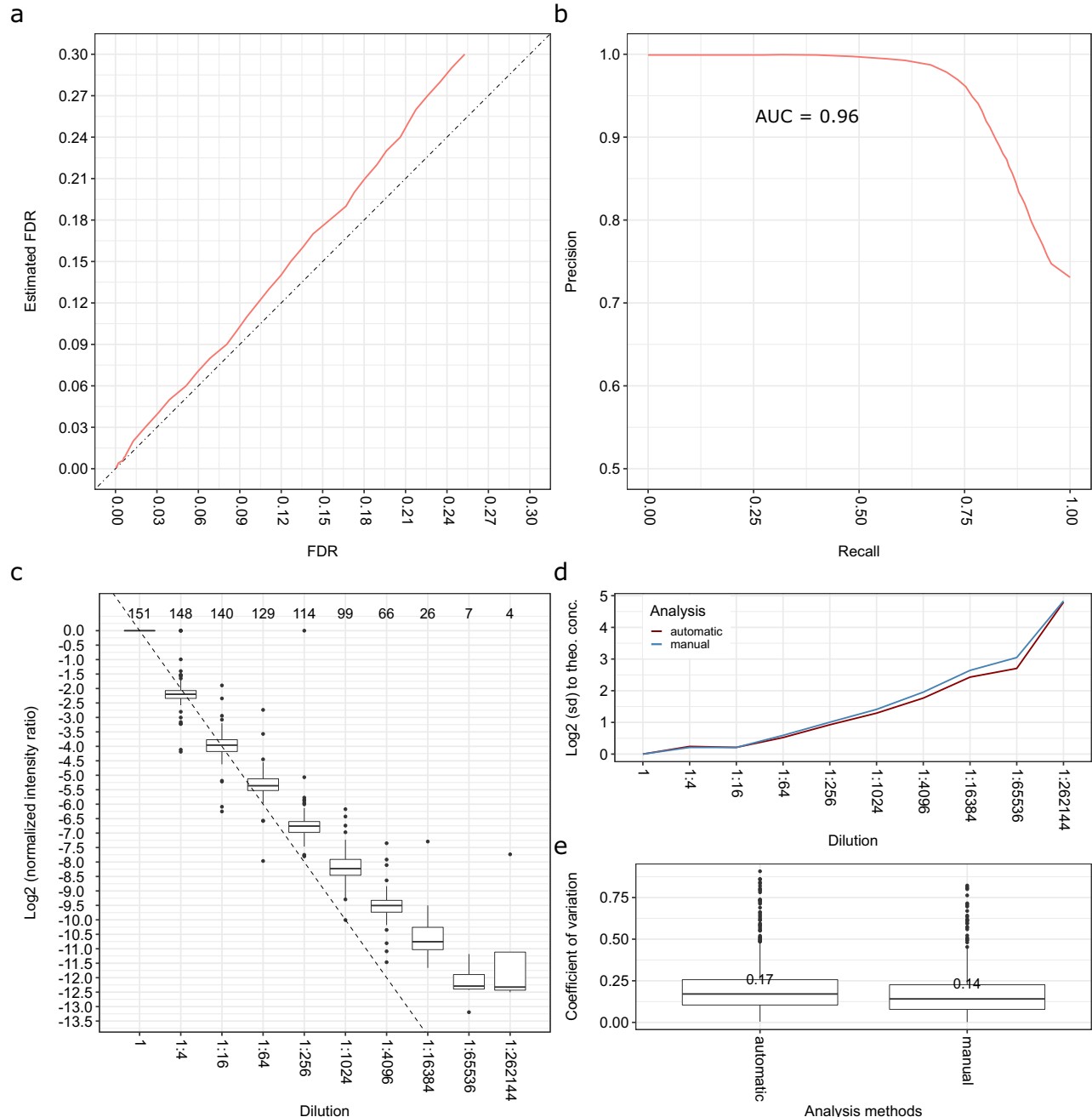

**Fig. 3 Identification accuracy and quantification of DIAMetAlyzer on the pesticide spike-in dataset. a** Estimated FDR versus FDR from the ground truth data. **b** Precision-Recall curve with the area-under-the-curve (AUC = 0.96). **c** Normalized intensity ratio over the dilution series. The dashed line indicates the expected fourfold difference to the next dilution. The x-axis (top): The number of metabolites found in the specific dilution at a 5% FDR cutoff. More than half of the initial metabolites could be detected at half of our dilution series (1:1,024). **d** Difference in mean standard deviation regarding the theoretical concentration of the automatic and manual analysis. **e** Median coefficient of variation (CV across three technical replicates for the automatic and manual analysis (CV < 20%)). For **c**, **d**, and **e**, only metabolites detected in triplicates and below a 5% FDR threshold were analyzed and only true positives were considered in the case of panel **e**. The box plots in **c** and **e** indicate median, 25th and 75th percentiles (middle line, Q1 and Q3 within the box, respectively), including 1.5x interquartile range whiskers and outliers (single points outside this range).

number of differences (220 differentially expressed features) using our identification-free pipeline, almost doubling the number of differentially expressed features.

**Biomarker detection**. Next, we analyzed the differentially expressed features to identify individual compounds that could serve as biomarkers of AMD or be involved in disease etiology (Supplementary Figs. 15–17). We found major differences

between control and patients in compounds associated with the classes glycerophospholipids, organic heterocyclic compounds, sterol lipids, fatty acids, amino acids, and dipeptides. Carnitines and their metabolites are mainly involved in fatty acid metabolism. Oleoylcarnitine ($P_{CNV} = 0.002$, $P_{PCV} = 0.01$), as well as L-Palmitoylcarnitine ($P_{PCV} = 0.02$), are upregulated by around 1.5 times in contrast to the control in both or PCV, respectively. These findings are consistent with previous research suggesting alterations in the carnitine shuttle pathway in macular

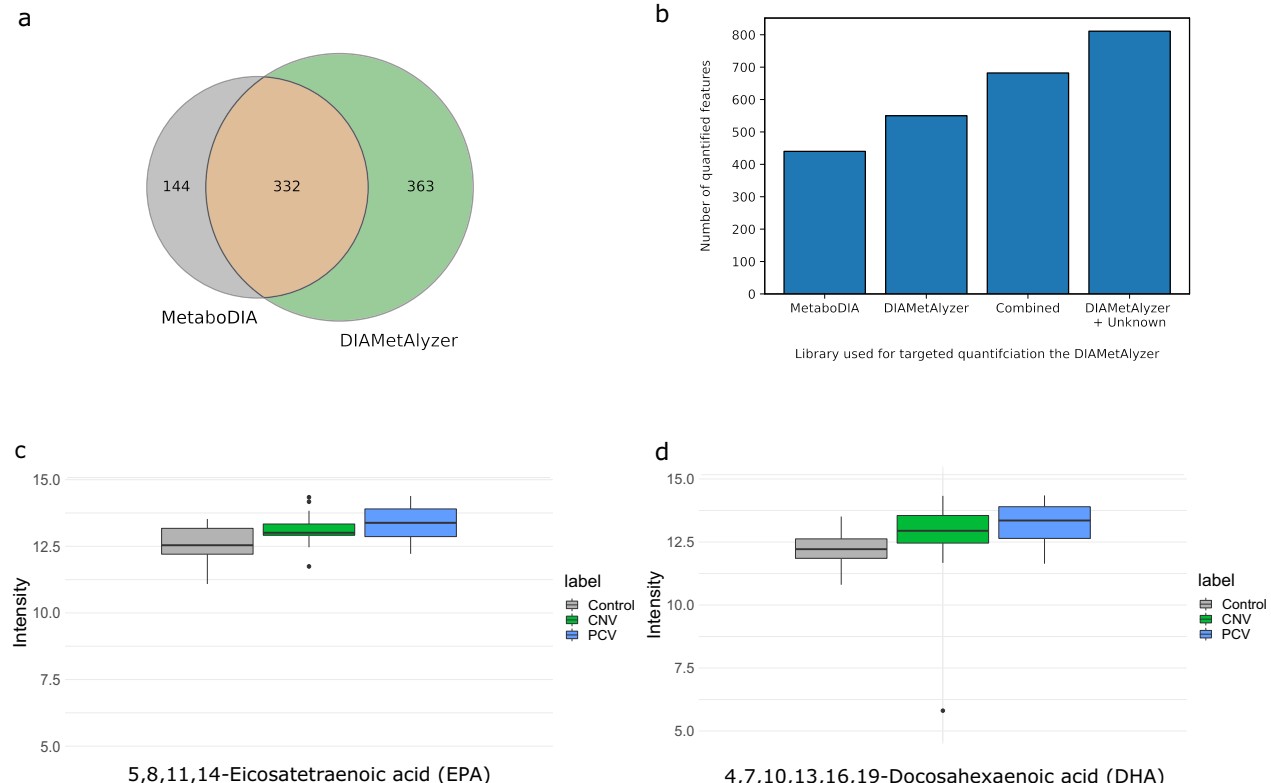

**Fig. 4 Analysis of serum samples of patients with AMD using MetaboDIA and DIAMetAlyzer. a** Comparison of the library generation of both tools based on features (molecular formula, adduct and retention time). 66% of the features overlap between the tools (DIAMetAlyzer: green, MetaboDIA: gray, Overlap: orange). **b** Number of quantified features using the various libraries in combination with the targeted extraction of DIAMetAlyzer. MetaboDIA, DIAMetAlyzer, the library of both tools (Combined), DIAMetAlyzer with the functionality to use known unknowns without prior MS1 identification additionally to the ones with identification (DIAMetAlyzer + Unknowns). **c** Significant deregulated compound 5,8,11,14-Eicosatetraenoic acid (EPA - C20H32O2 - based on putative identification) ($P_{CNV} = 0.04$; $P_{PCV} = 0.01$) with an increase in mean intensity of 1.4 and 1.7 times in contrast to the control. **d** Significant deregulated compound 4,7,10,13,16,19-Docosahexaenoic acid (DHA - C22H32O2 - based on putative identification) ($P_{CNV} = 0.008$; $P_{PCV} = 0.006$) with an increase mean intensity of 1.7 and 2.0 times in contrast to the control. **c, d** The identification of the compounds is based on MS1 accurate mass search and MS2 fragment annotation. Differential expression was assessed using limma with Benjamini-Hochberg correction. Box plots indicate median, 25th and 75th percentiles (middle line, Q1 and Q3 within the box, respectively), including 1.5x interquartile range whiskers and outliers (single points outside this range). Control: gray, $n = 20$ biologically independent samples, CNV (choroidal neovascularization): green, $n = 20$ biologically independent samples, PCV (polypoidal choroidal neovascularization): blue, $n = 20$ biologically independent samples.

degeneration[21]. Our findings suggest that Linoelaidylcarnitine ($P_{CNV} = 0.04$; $P_{PCV} = 0.03$), which showed a similar increase in addition to the others might be a potential biomarker for AMD. We found additional biomarker candidates using our identification-free pipeline at m/z 601.271468 and retention time 579 s ($P_{CNV} = 0.0002$; $P_{PCV} = 0.000008$) with an intensity increase of 5.2 and 5.7 in comparison to the control and at m/z 944.360964 and 311 s ($P_{CNV} = 0.005$; $P_{PCV} = 0.03$) with an intensity increase of 1.7 and 1.9 respectively. In addition to the findings above, we detected previously reported potential biomarkers associated with AMD, such as phenylalanine, hypoxanthine, tyrosine[22]. We found hypoxanthine levels were significantly increased in CNV ($P_{CNV} = 0.006$) by 3.9 times in contrast to the control, which affects the purine nucleotide cycle and can lead to apoptosis of photoreceptors[23,24]. In addition, gamma-Glutamylphenylalanine ($P_{CNV} = 0.002$; $P_{PCV} = 0.0006$), gamma-Glutamylisoleucine ($P_{CNV} = 0.01$; $P_{PCV} = 0.04$) and dityrosine ($P_{CNV} = 0.002$; $P_{PCV} = 0.03$) were deregulated in both patient groups. Increased serum gamma-glutamyl transferase (GGT) levels have previously been reported as risk factors for AMD[25]. This suggests that gamma-Glutamylphenylalanine with increased intensity by around 1.6 times in contrast to the control and gamma-Glutamylisoleucine (1.7 times increase) could

constitute useful metabolic markers for AMD. Additional biomarkers for AMD could lead to the development of predictive and or diagnostic models, allowing a deeper understanding of the disease and an earlier diagnosis, leading to a more timely treatment. In addition, it could allow the identification of potential therapeutic targets. As a validation, dityrosine, which we found increased around 1.7 to 2.4 times in contrast to the control, plays a role in oxidative stress and is associated with macular degeneration[26]. Interestingly, the significantly deregulated compounds 5,8,11,14-Eicosatetraenoic acid (EPA) ($P_{PCV} = 0.01$; $P_{CNV} = 0.04$) and 4,7,10,13,16,19-Docosahexaenoic acid (DHA) ($P_{PCV} = 0.006$; $P_{CNV} = 0.008$) demonstrate an increase of 1.4 to 2.0 times in contrast to the control. These have previously been associated with a reduced risk for neovascular AMD[27] (Fig. 4c, d). An explanation for this finding in the patients could be an Omega-3 fatty acids rich diet, which is often advised to AMD patients due to their anti-inflammatory properties[28,29]. The identification results of the differential expression analysis are based on putative identifications via MS1 accurate mass search and MS2 fragment annotation, corresponding to a level 3 identification[30]. Here, to reach a level 1 identification, additional experiments in follow up studies are necessary to validate the potential biomarkers.

## Discussion

It can be deemed as a limitation that DDA and DIA data have to be measured for an experiment. The main purpose of the DIA-MetAlyzer workflow is to perform accurate quantification in a targeted manner. Here, the DDA data - for example - reference standards would be measured once to construct the assay library. This library can then be reused for DIA data analysis measured with the same experimental setup. In a targeted setting, it is generally necessary to invest resources to build accurate assays in order to achieve high-quality targeted results. While DDA is generally biased towards high abundant analytes, this will not impact measurements of low complexity, such as pure standards. When building assay libraries from complex samples, the library will be biased towards highly abundant analytes. We suggest to counteract this bias by enhancing such assay libraries with reference compounds measured from pure standards.

DIAMetAlyzer uses SIRIUS for fragment annotation, so the limitations in terms of high-resolution instruments and molecular masses of SIRIUS apply to the workflow as well. High mass compounds can, in some cases, not be processed by SIRIUS in a timely manner. The user can set a threshold of 100 s (default) per compound, to restrict the runtime. As a reference, the assay library generated from 67 DDA samples, with prior MS1 identification took around 2.5 h using 10 cores (Intel(R) Xeon(R) Gold 6140 CPU @ 2.30 GHz). With allowing unknown features, it took around 12.5 h using 28 cores. The runtime of the complete KNIME workflow for the targeted pesticide mix experiment, using one core (Intel Core i7 @ 3.50 GHz), was 36 min. All runtime improvements of SIRIUS in the future will also impact the runtime of the workflow.

With the integration of the AssayGeneratorMetabo Node into OpenMS, we provide an easy-to-use solution for target-decoy assay library generation in OpenMS using the fragmentation re-rooting method[11]. Combining multiple feature detection methods similar to the combined MetaboDIA and DIAMetAlyzer library is not straightforward due to interoperability issues between the tools. For this purpose, we provide means to add decoys at the assay library level (DecoyGeneratorMetaboTool). For further details regarding the decoy methods on library level please see Supplementary Figs. 18 and 19. The so generated target and decoy assay library can then be appended to the one from the Assay-GeneratorMetabo. The combined library can be used in the DIAMetAlyzer workflow for the DIA data analysis.

In conclusion, we present an analysis workflow for metabolomics DIA data that introduces accurate control of the FDR. Our workflow is based on industry-grade computational libraries and workflow engines (OpenMS[6] and KNIME[31,32]) and builds on existing open-source software. It is possible to use the OpenMS command line tools and algorithms to build the workflow in any scripting environment, cluster, or cloud infrastructure. Our adaptive machine learning approach integrates the signal from MS1 and MS2 levels to optimally separate true signal from noise and provides a well-calibrated estimate of the FDR. In the past, without reliable estimates of precision in the reported data, DIA data was very difficult to analyse for mass spectrometry practitioners and available tools could result in vastly different results. Our introduction of a standardized workflow that utilizes a statistically well-calibrated FDR will allow practitioners to analyze and compare DIA data on equal footing. This extension allows for improvements in the reliability and robustness of metabolomics discovery. Importantly, our pipeline can be used in a targeted setting (quantifying known compounds) as well as in an untargeted setting (quantifying unknown compounds using their m/z patterns). In comparison to MS-DIAL, a software for untargeted deconvolution, we were able to detect almost twice as many compounds in the targeted setting. In comparison to MetaboDIA, a tool for consensus spectral library building for metabolomics data from DDA data, we were able to almost double the number of quantified features. In our analysis of a DIA dataset comparing serum of AMD patients to controls, our workflow allowed us to identify several previously undetected putative biomarkers for AMD. Specifically, the use of an experimentally specific DDA library based on reference substances allows for the accurate identification of compounds and markers from DIA data in low concentrations, facilitating biomarker quantification.

## Methods

**Chemicals**. Lyophilized human plasma was obtained from Sigma-Aldrich and prepared according to the supplied instructions (Sigma-Aldrich, Taufkirchen, Germany). LC-MS grade solvents were obtained from Sigma-Aldrich (Sigma-Aldrich, Taufkirchen, Germany). The Agilent LC/MS Pesticide Comprehensive Mix was obtained from Agilent Technologies (Agilent Technologies, Waldbronn, Germany).

**Sample preparation**. Benchmark samples were prepared by spiking different commercially available pesticide mixes (Agilent Technologies, Waldbronn, Germany) into human plasma metabolite extracts. Human plasma metabolite extracts were prepared by mixing one part of human plasma (Sigma-Aldrich, Taufkirchen, Germany) with three parts of precooled acetonitrile (ACN) (4 °C). After centrifugation at 15,871 x g at 4 °C for 15 min the supernatant was transferred, the solvent evaporated and the residue redissolved in 20% ACN at the original volume of the used plasma aliquot. This matrix was used to dilute the pesticide mixes in a dilution series according to Supplementary Table 2. Due to the molecular weight range of the pesticide mix, the different steps cover a concentration gradient of 5 orders of magnitude (Supplementary Fig. 5). For the preparation of the DDA data, each pesticide mix was diluted to 1 ng/μL with either solvent or plasma matrix. For the DIA data, a stock solution of all eight pesticide mixes in the plasma matrix was prepared with a concentration of 1 ng/μL.

**LC-MS/MS analysis**. The analysis was performed using a Nexera UHPLC system (Shimadzu) coupled to a Q-TOF mass spectrometer (TripleTOF 6600, AB Sciex). Separation of metabolites from the spiked human plasma metabolite extracts was performed using a UPLC BEH C18 2.1 × 100, 1.7 μm analytic column (Waters Corp.). The mobile phase was 0.1% formic acid in water (eluent A) and 0.1% formic acid in ACN (eluent B). The gradient profile was 5% B from 0 to 0.5 min, 100% B at 10 min for 3 min and 5% B at 13.5 to 16 min. A volume of 5 μL of the sample was injected. As indicated above different samples were measured in DDA and DIA/SWATH. MS settings were as follows: Gas 1 55, Gas 2 65, Cur 35, Temperature 500 °C, Ion Spray Voltage 5500 V, declustering potential 80 V Information Dependent Acquisition was used for the generation of assay libraries. The IDA duty cycle was 200 ms for MS1, 80 ms for MS2. The mass range of the TOF MS and MS/MS scans were 50–2000 m/z and the collision energy was ramped from 20–50 V or 50–80 V depending on the sample. SWATH acquisition was performed with one TOF MS survey scan (240 ms) followed by 8 SWATH scans (90 ms). The fragment ion window for SWATH was from 100 to 900 m/z. Here, variable windows were used, optimized on the plasma matrix using the SWATH Variable Window Calculator (SCIEX) (Supplementary Table 1).

**Analysis**. The initial DDA data processing for assay library generation was performed as shown previously, using Proteowizards qTofPeakPicker for centroiding and msconvert for the conversion to mzML[33]. Details regarding the conversion are available in the additional code repository (https://github.com/oliveralka/DIAMetAlyzer_additional_code/tree/master/convert_bash). The DIA data were converted using msconvert. The data was analysed using the described workflow with additional manual validation to acquire the ground truth data. Comparisons of ground truth data and additional statistical analysis was performed using python and R (https://github.com/oliveralka/DIAMetAlyzer_additional_code)[34]. For a visual inspection of representatives for DDA and DIA data, please see Supplementary Fig. 20.

**Workflow**. Our workflow is composed of steps for candidate identification, library construction, targeted extraction, and statistical validation (Fig. 1). Candidate identification. Data acquired using DDA is used as input for feature detection, adduct grouping, and accurate mass search. Feature detection is the process of annotating analytes based on their mass-to-charge, retention time, intensity, and charge[35]. Based on the feature space, adduct grouping is used to find possible adducts[36]. Annotated features and assigned adducts are then used by accurate mass search to extract potential compositions from a compound database. Library construction. Assay library generation is crucial for the targeted analysis of metabolomics DIA data[37]. In this context, we provide a tool called AssayGeneratorMetabo. It is implemented using the OpenMS C++ library[6]. The tool uses MS1 and MS2 spectral information and preprocessed feature information to

perform precursor correction and filtering based on the number of isotopic traces (data reduction). Afterward, feature mapping is performed to assign MS2 spectra to a specific feature. To ensure the validity of fragments, fragment annotation assigns fragments to their compatible metabolite substructures. In this approach, SIRIUS is used to assign compatible fragments to associated precursors[8]. From the fragment annotated spectra, $n$ highest intensity fragments are automatically extracted to generate potential transitions, which are used for assay library construction. At this stage, a targeted library is available. For the generation of the MS2 decoys, the fragmentation tree-based re-rooting method via Passatutto ensures the consistency of decoy spectra[11], which allows the estimation of a false-discovery rate later in the pipeline. The constructed assay library can be re-used to analyse a multitude of DIA/SWATH samples. Target-decoy assay libraries from other tools could also be used for the next step of the pipeline. Targeted extraction. The target-decoy assay library is used to analyse DIA/SWATH data. Targeted extraction involves chromatogram extraction and peak group scoring. This step is performed using a metabolomics-extended version of OpenSWATH[5], a well-established workflow commonly used in proteomics enabling the targeted analysis of DIA data. Statistical validation. FDR estimation is performed using the target-decoy library in combination with PyProphet[13–15], which was extended to deal with compound information. In PyProphet the OpenSWATH results were merged, scored on MS1 and MS2 levels using the metabolomics score filter and exported using the export-compound function. The pipeline is available as KNIME[31,32] workflow using OpenMS[6], pyOpenMS[38], SIRIUS[8], and Passatutto[11]. For further details, see Supplementary Fig. 21. The workflow is available in the OpenMS Tutorials (https://github.com/OpenMS/Tutorials) and on the OpenMS website (https://www.openms.de/comp/diametalyzer/). Additional details about the workflow can be found in the supplementary information.

**Assay library generation**. For assay library generation, the tool AssayGeneratorMetabo was implemented in C++ using the OpenMS Library[6]. It uses spectra information (.mzML) and preprocessed feature information (.featureXML). First precursor m/z and intensity are reannotated, then preprocessing such as filtering based on the number of isotope traces can be performed. Afterward, a feature mapping is used to assign a precursor and its MS2 spectra to a specific feature. After meta-information extraction, fragment annotation is performed using SIRIUS4[8]. From the annotated spectra, $n$ transitions are extracted based on a minimum/maximum intensity threshold. The same metabolite and adduct combination may be found multiple times in one sample using accurate mass search, for example, due to experimental reasons, such as column saturation or isobaric metabolites. Currently, the ambiguity is resolved by using the spectrum with the highest precursor intensity. The constructed target library can be exported in various formats (tsv, traML, pqp).

**Decoy generation**. The fragmentation tree-based method from Passatutto[11] was used for decoy generation. The fragmentation trees were acquired using fragment annotation via SIRIUS4[8]. The SIRIUS4 tree format had to be parsed into a Passatutto compatible format. After re-rooting, the decoy spectra were used to extract transitions. For overlapping transition and decoy transition masses after extraction, a -$CH_2$ mass was added to the overlapping decoy transition. To ensure the same number of targets and decoys, if re-rooting of the tree failed or the fragments were similar to the target ones, -$CH_2$ was added to the original fragment masses as a fallback mechanism to ensure the generation of a decoy. These fallbacks were used in around 13% and 5% of the cases, respectively (Supplementary Figs. 18 and 19). Afterward, the $n$ highest intensity peaks were extracted to use in the target-decoy assay library. On MS1 no additional decoy was generated.

**Manual validation**. The assay library was converted to a transition list using an in-house script (https://github.com/oliveralka/MetaboAssayLibToSkylineTransitionListConversion). The manual validation was performed using Skyline (19.1.0.193)[16]. Using default settings unless specified differently. The following transition settings were used. Fragments and precursors were used with the adducts ([M + H][M + K] [M + Na]) and all matching transitions were automatically selected. The instrument was set to 10 m/z (min) and 900 m/z (max) and a retention time from 0 to 16 min. MS1 filtering (up to 3 isotope traces) and MS/MS DIA with custom SWATH windows (Supplementary Table 1) were used. Scans within 2 min of MS/MS IDs were used.

**Assessment of the FDR calibration**. We annotated each peak group from our assay library manually. Here, a visual inspection was performed of the peak groups' presence, co-elution, and chromatographic shape. A true positive peak group is present if the precursor and transitions are properly co-eluting and show a chromatographic profile and the peak group is aligned within the dilution dataset (decreasing intensity along the dilution series). If the peak group is not of high quality (i.e., noise), it was excluded from the ground truth. Next, the FDR calibration was assessed by comparing the manually validated peak groups with those automatically detected. We constructed a confusion matrix for a predicted FDR threshold from 0.1% to 30% FDR. The confusion matrix reveals how many true and false hits we have detected based on the ground truth. We report a false positive when our software found a peak group where none was manually

annotated or if the retention time deviation was higher than 5 s. From the manual annotation, we compute the true false discovery rate: $FDR = FP / (FP + TP)$. Finally, the true FDR was compared to our estimated FDR using DIAMetAlyzer to assess its calibration. In addition, the matrix was used to determine other metrics such as precision and recall.

**Comparison with MS-DIAL**. The comparison between tools was based on the MTBL1108 dataset using MS-DIAL (Version 4.60). The data was preprocessed, and the assay library was converted to a spectral library. Results were filtered by available MS2 reference and were then compared to the ground truth and the 5% FDR filtered DIAMetAlyzer results. Please see the supplementary information for further details.

**Comparison with MetaboDIA**. The comparison between tools was based on the MTBLS417 dataset using MetaboDIA (Version 1.3) and a DIAMetAlyzer OpenMS development version (14f627e). Data was preprocessed[39–42]. Libraries were generated by both tools, with identification based on an accurate mass database including HMDB[18] (4.0) and LIPIDMAPS[19] (092020). All libraries were used for targeted extraction. Furthermore, statistical validation was performed and the results were reassessed based on chromatographic retention time alignment[43]. In the following, features with an FDR below 0.05 and the highest scoring peak group (rank 1) were used for post-processing analysis. The identification of the top significant features was assessed using MASST Search[44] (Supplementary Table 4). Please see the supplementary information for further details.

**Reporting summary**. Further information on research design is available in the Nature Research Reporting Summary linked to this article.

## Data availability
The spike-in benchmark dataset is publicly available in MetaboLights under accession code MTBLS1108. Comparison with MetaboDIA was performed using publicly available data MTBLS417. Databases used were HMDB 4.0 [https://hmdb.ca/], LIPIDMAPS (092020) [https://www.lipidmaps.org/].

## Code availability
OpenMS as open-source software is distributed under a BSD three-clause license and is available on Github (https://github.com/OpenMS/OpenMS). Additional code for re-analysis of the pipeline can be found on Github (https://github.com/oliveralka/DIAMetAlyzer_additional_code) [Version 1.1: https://doi.org/10.5281/zenodo.5913236][34].

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

## Acknowledgements

O.A. and O.K. acknowledge funding from 1019 the German Ministry of Research and Education (BMBF) under the grant (project 1020 031A5.035.0 A, de. NBI/CIBI, O.A., O.K.). We acknowledge support from the Open Access Publishing Fund of the University of Tübingen (O.A., O.K.). This project was funded by the Government of Canada through the CIHR (419634, P.S., H.R.) and NFRF (CRCC, 2018-01295, P.S., H.R.) as well as NSERC (CREATE 528163-2019, P.S., H.R.) and the University of Toronto. Also, we would like to thank Timo Sachsenberg, Tjeerd Dijkstra and Goerge Rosenberger for the support, discussions and additional input concerning PyProphet scoring. Furthermore, we would like to thank Hyungwon Choi, for our discussions related to MetaboDIA and MetaboKit.

## Author contributions

O.A., H.R., O.K. conceived the project. All authors supplied ideas to the experiment design. M.W. and K.K. performed sample preparation and data acquisitions. O.A. developed the method and performed the data analysis. P.S. supplied additional experiments, discussions and helped in preparing the publication. H.R. and O.K supervised the project. All authors discussed the results and edited the paper.

## Funding

## Competing interests

The authors declare no competing interests.
