## [Peer Review File · Nature Communications]

Reviewers' Comments:

Reviewer #1:

Remarks to the Author:

The submission describes a workflow for analyzing metabolomics experiments generated with data-independent acquisition MS. The workflow has two key elements: the library/assay generation including a decoy DB, and the DIA data processing pipeline with FDR-controlled feature detection. This contribution addresses a massive need of the metabolomics community for more sensitive, better, faster, etc. workflows for DIA-type of analyses. The presented workflow eclipses any previous efforts by transferring concepts and tools that transformed the field of proteomics. In addition, the workflow is available as a command-line tool and KNIME pipeline.

Technically, DIAMetAnalyzer constitutes a major leap. The manuscript, however, offers ample room for improvement to increase impact and readability.

(1) DIA in metabolomics

The major problem of this submission is that (in the introduction) it advocates for the advantages of DIA over DDA, but all results compare DIA vs. DIA. Fundamental differences exist between metabolomics and proteomics. They are rooted in fragmentation behaviour, structural heterogeneity, the magnitude of changes, etc. If DIA hasn't been widely adopted in metabolomics, it is not because of bottlenecks "in the creation of assay libraries and the processing of XICs". The reason is that thus far, nobody could convincingly show that DIA offers practical benefits over the canonical approach that adopts MS1-XICs for quantification and DDA-MS2 for identification.

By focusing solely on comparing DIAMetAnalyzer with other DIA software, this work loses attractiveness. The authors miss out on crucial experiments to make a strong case for DIA over DDA in metabolomics. Fundamental questions remain open: is LOD/coverage/accuracy of MS2-XICs in DIA better than classical untargeted MS1-XICs-centric workflows? Is it better than targeted workflows (e.g. <https://doi.org/10.1038/s41592-021-01307-z>)? What is the fraction of features identified by non-DIA workflows that is not captured by the DIA workflow? Regardless of the results, addressing such questions would set a true milestone for the use of DIA in metabolomics.

(2) Introduction

The introduction builds on outdated and wrong stereotypes. It needs to be corrected. For example, the text states: "Untargeted approaches use data-dependent acquisition (DDA), which selects a large number of metabolites for fragmentation in a data-driven (partially stochastic) manner. While this sampling allows for the detection of more compounds, its stochastic nature results in low reproducibility, an increased fraction of missing data, and reduced quantification accuracy". However:

(*) The selection of precursors in DDA is deterministic, not stochastic (ranking by intensity plus filters on charge, slopes, exclusion lists, etc). The stochasticity that was common 15 years ago when attempting to collect MS2 data for all peptides in shotgun proteomics has never been a significant issue in metabolomics. These days, the problem is effectively overcome by iterative DDA-type of scans (i.e. Acquire-X) or the numerous recent developments that allow for smarter precursor selection (i.e. Thermo, MaxQuant.live)

(*) Sampling doesn't increase the number of detected compounds, not in DDA/metabolomics that relies on MS1 for detection.

(*) Quantification accuracy is not affected by precursor selection. It is affected mainly by the number of points across a XICs peak trace, which is an issue of DIA runs.

Moreover, it would be beneficial to also enumerate the key challenges of DIA metabolomics. The only comment is on "creation of assay libraries and automatization of XICs processing" (line 53), but this is a minor aspect.

(3) FDR

FDR is a crucial element of the workflow but requires more explicit explanations. Reading through the text, I had the impression that the authors assume that readers know how DIA, FDR, etc. are addressed in proteomics. They tend to cite software (e.g. pyProphet, Passatutto) without describing the underlying task.

Importantly, the main text should clarify what the "discovery" in FDR stands for. What is, in practical terms, a false peak? How are the numbers of FPs and TPs in Figure 2a determined? It is

stated that the ground truth is manual annotation in Skyline, but probably only related to the pesticide peaks. How does the same analysis work in the case of the AMD study without spike-ins? How would such numbers be tested in a different experiment?

As a sanity check for FP/TP, it would be interesting to know if dropping the FDR_DIAMetAnalyzer filter (5%) affects the number of differentially expressed features in the AMD experiment. If the filtered features are only "false peaks", the number of features identified by LIMMA should be roughly constant.

(4) The benchmarks in Figure 3 need improvements.

Sensitivity (Figure 3c): Expressing LODs as dilution factor is non-sense. The LOD must be reported as amount (mol) or a concentration (Mol). Moreover, it is odd to aggregate the LODs of all pesticides in a unique value. The LOD differs across the tested pesticides and there is no reason to take an average. It also seems arbitrary to emphasize the point with 99 detections: why not 114 or 66? To demonstrate performance, the LOD of each pesticide should be determined and compared to that of MS1-XICs quantification (as in DDA) using formal criteria ($S/N > 10$ or so) instead of manual.

Precision: There is an elephant in the room, and Figure 3e does not address it. One would expect striking differences in precision between DDA and DIA because of major differences in scan rate and points per peak. The traces in Supp Fig 11 and 12 are neat examples that call for interpolation/resampling. To properly demonstrate precision, the CV of each pesticide and dilution should be determined and compared to that of MS1-XICs quantification. In addition, Figure 3c suggest a drift in accuracy, i.e. away from the dashed line. Please elaborate on this aspect, i.e. by a similar analysis of accuracy as proposed about for CV.

(5) Library construction

I understand that the library produced by DIAMetalyzer is larger than for MetaboDIA (Figure 4), but I had the impression that the authors recommend combining them, or maybe not. The main text is unclear on the topic. I had to read four pages of the supplement to learn more. The conclusion is that it is indeed better to combine as there is no obvious drawback, but all very much depends on the parameters used to identify features in the DDA experiment. Maybe it is sufficient to use the feature lists from other algorithms (next to openMS), but then use SIRIUS and Passatutto for T/D library generation. Frankly, the manuscript in its current form doesn't speak for a robust library generation workflow. I urge the authors to provide more concrete guidance in the main text.

Recovery is only reported for the pesticides (in Figure 2). These are a simple target because they are massively concentrated and have characteristic MS2 spectra. What about the library for the AMD study? The reported number of 811 features seems very limited for this type of sample. There must be a bottleneck somewhere. A more thorough analysis is necessary to assess the capacity of the workflow to build assays for a normal sample. How many MS1 features were found? How many features map to a DB entry? For how many was it possible to obtain an informative MS/MS? For how many was it possible to generate an assay? Please elaborate.

I don't know what to make of the simulation done with pesticides and NIST 17 (lines 135-144). As the authors state, libraries are experiment specific (line 64). The proposed simulation seems very far away from a real sample (i.e. the AMD study). Is it of any relevance to prove general quality of libraries? How can one verify the quality of the library in a more realistic setting, i.e. considering also dark matter, retention time, features that are not in the library, etc.?

As for Supp Figure 3B, it is not clear why 3 transitions should be preferred. The results for 25ppm/25ppm are almost identical between UIS1 and UIS3.

Minor points:

- Text structure: the text lacks a clear organization. Adding Sections headings would be a start.
- The limitations of DIAMetAnalyzer should be moved from the supplement to the main text.
- Line 18: "meaningful knowledge" > "meaningful data"
- Line 111: "only high quality assays were included in the library" > how is quality defined?
- Line 193-195: the sentence is incomplete.
- Line 428: what is the "peak group level"?
- LIPID MAPS > LIPIDMAPS

Reviewer #2:

Remarks to the Author:

Sorry for the long wait - I was swamped. The editor did a wonderful job following up with me and really tried to get a timely review out of me :). One of the main reasons that untargeted metabolomics analysis cannot be truly used at scale is because, unlike proteomics or sequencing or other omics technologies, there are no good ways to control FDR. I had reviewed this paper for NMeth and thought it was a good paper then and think that this is a good paper now. I -and I am sure the authors even more- was disappointed in the decision at NMeth as one of the main reasons for rejection was that a reviewer picked one inaccuracy of an annotation (which could have been textually addressed) and extrapolated that the method is incorrect. That was a complete misunderstanding by the reviewer of FDR and the role FDR plays in metabolomics analysis. The whole point of FDR estimation is so one can judge how many inaccuracies they are willing to accept before including them in the downstream analysis and biological interpretation. By definition of FDR, there will be inaccuracies. Proteomics, sequencing and many other OMICS fields utilize and rely on such FDR estimators. I am convinced that this paper will spur the development of other FDR methods and over time. Thus the field will collectively, and as a community, learn how to best do this in metabolomics. In short, this is an important paper, it's really the only method for the type of metabolomics data (DIA) that currently exists and thus is a key milestone in my opinion. It is also a key milestone as it will be foundational to push the ability to perform DIA metabolomics at the population scales - in which data independent analysis will play a key role in the future as it is more reproducible than DDA, the only other data format for which FDR exists. The impact of this paper will bear out long-term not short term as 99% of metabolomics labs don't know how to use or think about FDR and this requires continuous education of the entire community. Perhaps the authors should get together with many other top labs and write a review/perspective on the importance of FDR in metabolomics. The work is nicely done and thanks for providing such a great resource to the community, especially now it's not only available as KNIME functionality!

Response to reviewers' comments

Reviewer #1:

(1) DIA in metabolomics

The major problem of this submission is that (in the introduction) it advocates for the advantages of DIA over DDA, but all results compare DIA vs. DIA. Fundamental differences exist between metabolomics and proteomics. They are rooted in fragmentation behaviour, structural heterogeneity, the magnitude of changes, etc. If DIA hasn't been widely adopted in metabolomics, it is not because of bottlenecks "in the creation of assay libraries and the processing of XICs". The reason is that thus far, nobody could convincingly show that DIA offers practical benefits over the canonical approach that adopts MS1-XICs for quantification and DDA-MS2 for identification.

By focusing solely on comparing DIAMetAnalyzer with other DIA software, this work loses attractiveness. The authors miss out on crucial experiments to make a strong case for DIA over DDA in metabolomics. Fundamental questions remain open: is LOD/coverage/accuracy of MS2-XICs in DIA better than classical untargeted MS1-XICs-centric workflows? Is it better than targeted workflows (e.g. <https://doi.org/10.1038/s41592-021-01307-z>)? What is the fraction of features identified by non-DIA workflows that is not captured by the DIA workflow? Regardless of the results, addressing such questions would set a true milestone for the use of DIA in metabolomics.

We would like to clarify that we do not aim to compare DIA with DDA in our manuscript, our introduction attempts to make the reader aware of major differences between the methods. Other studies already compared DDA with DIA, noting both advantages and disadvantages for each method; both approaches have their pros and cons, and the choice of method will depend on the goal of the study. This is why our work is motivated by integrating both techniques, thus leveraging their respective advantages. A recent study by Guo and Huan¹ addresses some of the questions raised by the reviewer above: In their study, DIA improves quantification, as well as precision and shows significant advantages in MS2 spectral coverage over DDA, whereas DDA yields significantly better MS2 spectrum quality. We aim to combine the best of both approaches by deriving libraries based on high-quality DDA MS2 spectra with few interferences and then subsequently use DIA to perform quantification, exploiting the improved MS2 coverage and superior quantification performance of DIA.

In order to clarify our approach that aims to combine DDA and DIA by playing to the strength of each method, we have now further expanded the text in the introduction to reflect and clarify this. We hope the reviewer will find the revised text more appropriate to capture the essence of our study.

For further points on the introduction raised by the reviewer, please see our replies to the second point below as well.

(2) Introduction

The introduction builds on outdated and wrong stereotypes. It needs to be corrected. For example, the text states: "Untargeted approaches use data-dependent acquisition (DDA), which selects a large number of metabolites for fragmentation in a data-driven (partially stochastic) manner. While this sampling allows for the detection of more compounds, its stochastic nature results in low reproducibility, an increased fraction of missing data, and reduced quantification accuracy". However:

(*) The selection of precursors in DDA is deterministic, not stochastic (ranking by intensity plus filters on charge, slopes, exclusion lists, etc). The stochasticity that was common 15 years ago when attempting to

collect MS2 data for all peptides in shotgun proteomics has never been a significant issue in metabolomics. These days, the problem is effectively overcome by iterative DDA-type of scans (i.e. Acquire-X) or the numerous recent developments that allow for smarter precursor selection (i.e. Thermo, MaxQuant.live)

We agree with the reviewer that the introduction should have been phrased more clearly. We have rewritten the introduction to more clearly define the scope of the manuscript. We aim to present substantial improvements and automation in the field of targeted DIA/SWATH analysis, however, a direct comparison of DIA vs. DDA is not in the scope of our manuscript. Other recent studies have done so, for example Guo and Huan compare DDA and DIA performance for metabolomics analyses (see also above). They find that the main problem of DIA is that a lot of false-positive metabolite features are observed in the low signal-to-noise ranges of DIA results. We want to tackle this problem with our automated approach by introducing a well-calibrated method for estimating and controlling the false-discovery rate (FDR).

Based on the reviewer's criticism regarding the stochasticity, we removed the corresponding section in the introduction. We wanted to express that, even though the precursor selection algorithms are certainly deterministic, the underlying stochasticity of the data, i.e., the variation between technical replicate in retention time and intensity, leads to different precursors being selected between technical replicates of the same sample. This is also what Guo and Huan observed in their comparison with DDA providing lower MS2 spectral coverage than DIA. However, this discussion is not central to our approach, so we removed it to avoid misunderstandings.

Main text:

Untargeted approaches often use data-dependent acquisition (DDA), which selects a large number of metabolites for fragmentation in a data-driven manner, based on the precursor selection. Data-independent acquisition (DIA) cycles through a series of predetermined mass ranges (DIA or SWATH windows) to acquire a high-resolution MS2 spectrum, thus boosting reproducibility by sampling the entire mass range. This allows for the systematic, unbiased acquisition of fragmentation spectra, at the cost of acquiring highly multiplexed spectra since the mass isolation range for each DIA window is generally larger than for other methods. A comparison of DDA and DIA data acquisition revealed that DDA excels in MS2 spectrum quality, whereas DIA shows a better performance in quantitative precision and MS2 spectrum coverage¹. A major challenge of DIA for the field is the measurement of multiplexed spectra, which are considered lower quality.

(*) Sampling doesn't increase the number of detected compounds, not in DDA/metabolomics that relies on MS1 for detection.

We agree with the reviewer - we have removed that statement from the introduction.

(*) Quantification accuracy is not affected by precursor selection. It is affected mainly by the number of points across a XICs peak trace, which is an issue of DIA runs.

We agree with the reviewer that the number of points across an XIC trace is an important factor in quantification accuracy and that precursor selection does not affect quantification accuracy but rather reproducibility. Other factors affecting quantification accuracy are spectral acquisition time, intra-scan dynamic range, signal-to-noise, matrix complexity and separation power (i.e. interference by isobaric or almost isobaric compounds). Some of these factors will be better in DIA, others will be worse. A direct comparison of quantification accuracy has to be experimentally determined and is out of scope in our manuscript, however recent studies show that DIA has at least comparable if not superior quantification accuracy compared to DDA methods¹. In our analytical methods, with cycle times of around 1 s (in our case), we generally obtain good coverage of elution profiles. DDA methods do not necessarily provide shorter cycle times (and better elution profile sampling), rather this will depend on the specific analytical method chosen. We agree, however, that this is something to keep in mind when optimizing the acquisition methods. It is hard to compare the influence of these factors theoretically, we have therefore removed the corresponding text in the introduction.

Moreover, it would be beneficial to also enumerate the key challenges of DIA metabolomics. The only comment is on “creation of assay libraries and automatization of XICs processing” (line 53), but this is a minor aspect.

We agree and we have expanded the introduction on this point. The revised manuscript also mentions the detection and quantification of noise peaks/peak groups with low intensities as well as the quality of the resulting MS² spectra, and the need for more advanced data processing as key challenges for DIA-based metabolomics.

Main text:

While both the creation of assay libraries for DIA analysis and the processing of XICs has been automated in other fields², this is not the case in metabolomics. Additionally, a main challenge in targeted metabolomics is the detection of false positive metabolic features in the low signal-to-noise ranges of DIA results that are unable to be filtered¹.

Here, we present a novel workflow based on the targeted strategy which solves these issues, first integrating a complete end-to-end pipeline including assay library generation into a widely used software suite (OpenMS³) and secondly implementing a novel procedure to estimate robust and accurate false-discovery rates (FDRs) for DIA metabolomics. Our DIAMetAlyzer software combines DDA and DIA metabolomics data by deriving libraries based on high quality DDA MS² spectra with few interferences and then subsequently uses DIA to perform quantification, exploiting the improved MS² coverage and superior quantification performance of DIA¹.

(3) FDR

FDR is a crucial element of the workflow but requires more explicit explanations. Reading through the text, I had the impression that the authors assume that readers know how DIA, FDR, etc. are addressed in proteomics. They tend to cite software (e.g. pyProphet, Passatutto) without describing the underlying task.

We have recognized that our presentation of the underlying concepts was rather brief and we have now expanded the description of these concepts for the benefit of the reader.

Additions to the main text:

Targeted extraction. The assay library is then used to analyze DIA data. Targeted extraction involves chromatogram extraction and peak group scoring in OpenSWATH² (Supplementary Fig. 20). We modified OpenSWATH to support targeted extraction of metabolomics data, which are included in the latest release of OpenSWATH. Here, chromatograms in a user-specified retention time window are extracted from the DIA data based on the transitions entries in the assay library. They encompass precursor and its isotope traces as well as the specified MS2 fragment traces. All extracted traces are grouped in so-called peak groups, which represent a possible analyte with MS1 and MS2 traces. For each peak-group a score matrix based on different scores, such as co-elution and chromatogram shape. A detailed description of the OpenSWATH scores can be found in the original publication.

Importantly, the main text should clarify what the “discovery” in FDR stands for. What is, in practical terms, a false peak?

Addition to the supplementary material:

Targeted extraction, scoring and FDR estimation

The false discovery rate (FDR) was introduced, handling the results of the increasing amounts of genomics data. It is the expected ratio of false-positive classifications (false discoveries) to the total number of positive classifications.

$$\text{FDR} = \text{FP} / (\text{FP} + \text{TP})$$

In this context, “discovery” is a statistical term rooted in the context of positives and negatives. In our context, the “positives” would be true quantification events (eg a signal in the XIC that corresponds to the analyte of interest) while the “negatives” would be false quantification events (e.g. a signal / region in the XIC that does not correspond to the analyte of interest and is either chemical noise or belongs to another analyte). In original terminology as introduced in 1995, the “discovery” stands for the items that are labelled as “positive”, and hence could be true positives or false positives, in a gene expression experiment as the genes that you label as differentially expressed⁴.

In 2007, Elias et al. introduced the concept of target-decoy FDR in mass spectrometry-based proteomics⁵, where it is used to distinguish correct from incorrect peptide identifications.

Unfortunately, creating plausible decoys in metabolomics is not as straightforward since methods like shuffling or reversing a sequence do not work in metabolomics. However, target-decoy based FDR estimation was introduced recently for large scale untargeted metabolomics studies^{6,7}. Scheubert et al. presented methods to ensure the consistency of the decoy spectra by using

fragmentation tree re-rooting. First, a fragmentation tree is constructed for the original spectrum identification assigning fragments compatible with the metabolite substructures. In a second step, this fragmentation tree will be re-rooted. A new root is chosen, leading to tree rearrangements by shifting the fragmentation reaction order. The results are new potential fragments based on the original metabolite. The generated decoys should have a similar probability of occurring in the sample and could represent the same metabolite - a decoy. The FDR for hits in a spectral library database can be estimated with this method.

In the targeted setting, the peak group FDR estimation works differently. Experimental specific targets and decoys are added to the assay library (prior knowledge database) used for targeted extraction. For available targets and decoys, peak groups are extracted and scored (Supplementary Fig 20). A discriminant score (d-score) distribution is computed using a linear discriminant analysis based on the available subscores, and statistical error estimates are derived by fitting a null distribution⁸⁻¹⁰ (see suppl Fig. 8).

Supplementary Fig. 20 | Targeted extraction and scoring Three transitions belonging to an analyte are extracted from DIA MS/MS data in a defined retention time window. Peak group candidates are extracted as chromatograms, scored and validated.

In terms of FDR control, the importance or span of the FDR depends on the research question. For example, a low number of false positives in clinical biomarker identification is important (1% - 5% FDR). On the other hand, if the aim is to find new potential biomarkers, an FDR of 10% might still be valid since these must be further validated.

Addition to the main text:

Statistical validation. FDR estimation originated from the increasing amounts of data in the genomics field. It is the expected ratio of false positive classifications (false discoveries) to the total number of positive classification. The “discovery” stands for the items that you label as “positive”, and hence could be true positives or false positives, in the gene expression sense as the genes that you label as differentially expressed⁴. In 2007, Elias et al. introduced the concept of target-decoy FDR in proteomics⁵, where it is used to distinguish correct from incorrect peptide identifications. In the targeted field experimental specific targets and decoys are added to the assay library (prior knowledge database) used for targeted extraction. For available targets and decoys, peak groups are extracted and scored (Supplementary Fig. 20). We use semi-supervised learning to build a composite score (discriminant score) out of individual peak group scores and estimate q-values by fitting a null distribution using a version of PyProphet adopted to metabolomics⁸⁻¹⁰. To prevent overfitting, we chose a straightforward linear model (LDA) for target-decoy discrimination using peak group scores with a low cross-correlation, which resulted in an excellent performance on our benchmark dataset (Supplementary Fig. 8).

How are the numbers of FPs and TPs in Figure 2a determined? It is stated that the ground truth is manual annotation in Skyline, but probably only related to the pesticide peaks. How does the same analysis work in the case of the AMD study without spike-ins? How would such numbers be tested in a different experiment?

Validation of the proper FDR calibration requires a defined ground truth. The ground truth in our case is represented by the manual validation. We manually inspected each peak group from our assay library manually in Skyline. Basically, we validated whether the detected signal was plausible and did not have the appearance of noise. In the general case, as in the AMD study, this is a very difficult task and even an experienced validator may not always make the correct call. However, in our pesticide dataset, an annotator could use the additional information present in the dilution series to further validate their choice: if the peak intensity is decreasing with increasing dilution, the peak is most likely generated by a pesticide and not by the background sample matrix. This allowed our annotators to utilize orthogonal information to generate a high-quality ground truth dataset that can be used to now benchmark the FDR estimation of our algorithm (which does not have access to the orthogonal information of the dilution information). A true positive peak group is present if the precursor and MS2 traces are properly co-eluting and have a fitting shape (Fig. P1).

FDR calibration was then checked by comparing the manually curated ground truth peaks with the ones automatically identified by DIAMetAlyzer. For predicted FDR thresholds ranging from 0.001 to 0.3 we built a confusion matrix detailing how many true hits (same in the ground truth) and how many false hits we have detected (the hit was false if the retention time deviation was higher than 5s or if we reported a hit where there should not be one). From the annotation in the confusion matrix we can calculate the actual FDR based on:

$$\text{FDR} = \text{FP} / (\text{FP} + \text{TP})$$

Now we can compare the prediction with the actual calculated FDR to see how well the predicted FDR is calibrated. In addition, the confusion matrix can be used to assess other metrics such as precision and recall.

In other words, the assessment of the FDR calibration for other datasets (i.e. AMD) is very time consuming and cannot be done with the same level of confidence as with the pesticide dataset since it is not a spike-in dataset. We therefore cannot compare to known TP and FP numbers on that dataset, however our analysis on the pesticide dataset allows us to conclude that the FDR estimation is well calibrated and should produce accurate results on the AMD dataset as well.

Main text (Methods):

Assessment of the FDR calibration

We annotated each peak group from our assay library manually. Here, a visual inspection was performed of the peak groups' presence, co-elution, and chromatographic shape. A true positive peak group is present if the precursor and transitions are properly co-eluting and show a chromatographic profile and the peak group is aligned within the dilution dataset (decreasing intensity along the dilution series). If the peak group was not of high quality (i.e., noise), it was excluded from the ground truth. Next, the FDR calibration was assessed by comparing the manually validated peak groups with those automatically detected. We constructed a confusion matrix for a predicted FDR threshold from 0.1% to 30% FDR. The confusion matrix reveals how many true and false hits we have detected based on the ground truth. We report a false positive when our software found a peak group where none was manually annotated or if the retention time deviation was higher than 5s. From the manual annotation, we compute the true false discovery rate: $FDR = FP / (FP+TP)$. Finally, the true FDR was compared to our estimated FDR using DIAMetAnalyzer to assess its calibration. In addition, the matrix was used to determine other metrics such as precision and recall.

As a sanity check for FP/TP, it would be interesting to know if dropping the FDR_DIAMetAnalyzer filter (5%) affects the number of differentially expressed features in the AMD experiment. If the filtered features are only "false peaks", the number of features identified by LIMMA should be roughly constant.

Our FDR only estimates whether a given signal in the data maps to a targeted assay (i.e., a metabolite) and not whether a peak is differentially expressed. Therefore, decreasing the FDR threshold makes the filtering more strict and only very high quality peaks of high intensity will be retained, however differentially expressed metabolites may be found both in the high-intensity and low-intensity part of the dataset. By dropping the FDR for example from 5% to 1% we would remove on repeated instantiation of the same experiment 38 false positive peaks but also remove 430 true positive peaks. We would therefore expect to remove many true positive signals as well and cannot expect that the number of features identified by LIMMA would be constant.

As we showed in Figure 2a (see manuscript), if we filter by different FDR thresholds, we reduce not only the amount of potential false positive peaks but also the number of the true positives.

Fig. P2 | Peak group d-score density diagram The diagram shows the d-score density distribution for the target and decoy peak groups.

If we image the filter as a vertical line separating the d-score at 0, we would allow a huge part of the decoy population and probably a huge part of false positives in the analysis (Fig. P2 / Supplemental Fig. 8). Of course, the further the filter is shifted to the right, the smaller the proportion of probably false positive peaks will be, but there is always a chance that also true positive peaks are filtered out.

So in the case of limma, it is not necessarily true that the feature space will be static. Some of the true features will be removed by filtering at 1% instead of 5%, which are then also removed from the LIMMA analysis.

(4) The benchmarks in Figure 3 need improvements.

Sensitivity (Figure 3c): Expressing LODs as dilution factor is non-sense. The LOD must be reported as amount (mol) or a concentration (Mol). Moreover, it is odd to aggregate the LODs of all pesticides in a unique value. The LOD differs across the tested pesticides and there is no reason to take an average. It also seems arbitrary to emphasize the point with 99 detections: why not 114 or 66? To demonstrate performance, the LOD of each pesticide should be determined and compared to that of MS1-XICs quantification (as in DDA) using formal criteria ($S/N > 10$ or so) instead of manual.

We calculated the LOD using the unfiltered results and the definition of ($S/N > 10$) for each metabolite (Table PT1). The results were added as a table to the supplementary information. In our opinion, it is customary to aggregate data in a statistical fashion when more data is computed

than can be visually presented in a single graph (e.g., averaging our filtered and detected 99 compounds at half the dilutions series (1:1,024)). We reasoned that the number of pesticides at 50% of our dilution series filtered using a 5% FDR cut-off would indicate how many signals are still detectable. Therefore, we revised the main text to clearly state that this is the final concentration a compound was detected based on our 5% FDR filter. The LOD calculation based on the unfiltered results can be found in the supplementary information.

Changes to the main text:

To determine the quantification performance, the results were filtered using a 5% FDR threshold and normalized for each metabolite adduct combination by the intensity of their highest concentration. More than half of the initial metabolites could be detected at half maximal dilution (1:1,024), based on the last dilution step a metabolite was observed in (Fig. 3c, Supplementary Fig. 7). The limit of detection of the individual metabolites were assessed using the unfiltered results, based on an S/N threshold of 10 (Supplementary Table 3). Comparing the quantification of manual and automatic analyses, the precision of the automated method matches manual analysis and outperforms it in some dilution steps (Fig. 3d). In all technical replicates, the median coefficient of variation (CV) of non-normalized quantified signals was smaller than 0.2 (Fig. 3e).

Added to supplementary material:

Limit of detection

We calculated the LOD, using the unfiltered results and the definition of ($LOD = S/N > 10$) for each pesticide (Supplementary Table 3). We used the intensity at the lowest dilution the compound was still detected via targeted extraction as noise. From there we calculated the concentration based on the exact mass, the initial concentration of 1ng/μl and the corresponding dilution of the compound. Please see *lod.RMD* at https://github.com/oliveralka/DIAMetAlyzer_additional_code for details.

Suppl. Table 3 | Limit of detection of the individual pesticides

compound name	molecular formular	last SN over threshold	last dilution over threshold	LOD [fmol/μl]
Acephate	C4H10NO3PS	15	3	342
Sulfadiazole	C7H12N4O3S2	15	7	237
.				
.				
.				

Precision: There is an elephant in the room, and Figure 3e does not address it. One would expect striking differences in precision between DDA and DIA because of major differences in scan rate and points per peak. To properly demonstrate precision, the CV of each pesticide and dilution should be determined and compared to that of MS1-XICs quantification.

As discussed above in more detail, the direct comparison of DDA and DIA data acquisition is not within the scope of our manuscript. First, we did not measure the dilution series in DDA, so we cannot compare it directly. Second, a similar comparison between DDA and DIA for untargeted metabolomics has been performed previously¹. It concludes that each method has its strengths and weaknesses and that the optimal method depends on the experimental context. DIA was shown to positively impact quantitative precision and MS2 spectral coverage in metabolomics data compared to DDA. In general, DDA data does not necessarily have more points per peak than DIA/SWATH. The quantitative accuracy as well as the number of points per peak will depend on the experimental design and the acquisition time of MS1 and MS2 spectra. We would assume that it is possible to have roughly the same scan time for MS1 if the acquisition mode is not full-scan (MS1 only).

As proof of concept, we visually compared the DDA and DIA XICs for the highest concentration and chose two representatives (Fig. 3P). Both show a similar chromatogram shape on the MS1 level. At the MS2 level, the peak depends on the trigger time of the instrument (the chromatogram shape is based on the interpolation used in the Skyline visualization). For DIA MS2, the chromatogram is nicely visible and co-elutes with the MS1 chromatogram.

Supplemental Material:

We visually compared the DDA and DIA XICs for the highest concentration and chose two representatives (Supplementary Fig. 21). Both show a similar chromatogram shape on the MS1 level. At the MS2 level, the peak depends on the trigger time of the instrument (the chromatogram shape is based on the interpolation used in the Skyline visualization). For DIA MS2 the chromatogram is clearly visible and co-elutes exactly with the MS1 chromatogram.

Suppl. Fig. 21 | Examples for MS1 and MS2 XICs from DDA and DIA data. The study data representative pesticides Bifenazate (a) and Rimsulfuron (b) show similar MS1 XICs in DDA and DIA.

The traces in Supp Fig 11 and 12 are neat examples that call for interpolation/resampling.

We agree that the shown features detected in the AMD DDA data show interpolation/resampling (suppl. Figure 11c,d, and 12 a-c). We added these examples to give an overview of features uniquely detected by the different feature detection algorithms (XCMS/CAMERA and FeatureFinderMetabo). This does not represent the main population of features detected in the analysis. In the case of Figure 11c,d, the analytes are eluting at the beginning of the gradient and are somewhat cut-off. But they are still detectable by the XCMS algorithm. In the case of Figure 12 a-c, the feature intensity span in the file was roughly from 7,523 to 72,903,272. All features shown have a maximum intensity of 75,000 and can be deemed low intensity. We agree that they do not show an optimal chromatographic shape with around 6-8 peaks. Still, they are detected as valid features from the FeatureFinderMetabo algorithm. Figure 4P shows the chromatograms of

two mid-range intensity features and one high-intensity feature representing the main feature population in the dataset.

Supplementary Material:

The presented features in Figures 11 and 12 do not represent the main population of features detected in the analysis. In the case of Figure 11 c,d, the analytes are eluting at the beginning of the gradient and are somewhat cut-off. But they are still detectable by the XCMS algorithm. In the case of Figure 12 a-c, the feature intensity span in the file was roughly from 7523 to 72903272. All features shown have a maximum intensity of 75000 and can be deemed low intensity.

The features presented in Figures 11 and 12 do not show an optimal chromatographic shape with around 6-8 peaks. Supplementary Fig. 13 shows the chromatograms of two mid-range intensity features and one high intensity feature representing the main feature population in the dataset. Please be aware that due to different axis ranges in intensity and retention time, the figures are not directly comparable, but should give an indication of the feature population in the sample.

Suppl. Fig. 13 | Examples for mid and high intensity features detected in the DDA data. a,b) Mid intensity features with an intensity of around 2000000 were detected in both algorithms. c) High intensity feature with an intensity of 7000000. All three features were extracted from DDA data and are shown as representation of the chromatographic outline of the main feature population in the dataset.

In addition, Figure 3c suggest a drift in accuracy, i.e. away from the dashed line. Please elaborate on this aspect, i.e. by a similar analysis of accuracy as proposed about for CV.

We assume that the drift in terms of quantification values is due to reaching the limit of quantification/detection and we would like to point out that both our automated analysis as well as manual analysis of the same data suffer from this issue, we therefore conclude that this is not an artifact introduced by our algorithm.

(5) Library construction

I understand that the library produced by DIAMetAlyzer is larger than for MetaboDIA (Figure 4), but I had the impression that the authors recommend combining them, or maybe not. The main text is unclear on the topic. I had to read four pages of the supplement to learn more. The conclusion is that it is indeed better to combine as there is no obvious drawback, but all very much depends on the parameters used to identify features in the DDA experiment. Maybe it is sufficient to use the feature lists from other algorithms (next to openMS), but then use SIRIUS and Passatutto for T/D library generation. Frankly, the manuscript in its current form doesn't speak for a robust library generation workflow. I urge the authors to provide more concrete guidance in the main text.

With the *AssayGeneratorMetabo* Node in the DIAMetAlyzer workflow, we provide a robust, automated and reproducible way to generate target and decoy assay libraries based in DDA data. Combining the feature detection results (e.g., spectral libraries) from other tools has its benefits, but it is not straightforward due to interoperability issues between the tools from different sources and the used data formats. In most cases, individual solutions are necessary to convert the original format to the OpenSWATH assay library format. Then decoys can be generated using methods developed to work on assay library level (Supplementary Fig. 9). For this purpose, we provide the *DecoyGeneratorMetaboTool*. The so generated target and decoy assay library can then be appended to the one from the *AssayGeneratorMetabo* Node. The combined library can be used in the DIAMetAlyzer workflow for the DIA data analysis.

Supplemental Material:

The created *AssayGeneratorMetabo* Node in our DIAMetAlyzer pipeline provides a robust, reproducible, automated method to build the target-decoy assay library in the OpenMS ecosystem using the fragmentation tree re-rooting method⁶.

The combination of spectral/assay library information of other tools is not straightforward due to interoperability issues between the tools and their used data format.

In most cases individual solutions need to be provided/developed to convert the original format to the OpenSWATH assay library format. In the case of MetaboDIA we provide a script for the conversion (`convertSpectralLibrarytoAssayLibrary_1.4.py`).

The fragmentation re-rooting method (SIRIUS) can not be performed since the MS1 and MS2 spectra information needed is not available on assay library level. For that purpose, we investigated decoy generation methods on the assay library level, which are easier to use and still perform reasonably well. Here, we provide the *DecoyGeneratorMetaboTool* (https://github.com/oliveralka/DIAMetAlyzer_additional_code/blob/master/additional_tests_decoy_methods/TOOL/DecoyGeneratorMetaboTool_2.0.py). For further details regarding the decoy methods on library level please see Supplement Fig. 9. The so generated target and decoy assay library can then be appended to the one from the *AssayGeneratorMetabo* node. The combined library can be used in the DIAMetAlyzer workflow for the DIA data analysis.

Main text:

With the integration of the *AssayGeneratorMetabo* into OpenMS, we provide an easy-to-use solution for target-decoy assay library generation in OpenMS using the fragmentation re-rooting method⁶. Combining multiple feature detection methods similar to MetaboDIA and DIAMetAlyzer is not straightforward due to interoperability issues between the tools. For this purpose, we provide means to add decoys on the assay library level (*DecoyGeneratorMetaboTool*). For further details regarding the decoy methods on library level please see Supplementary Fig. 9. The so generated target and decoy assay library can then be appended to the one from the *AssayGeneratorMetabo*. The combined library can be used in the DIAMetAlyzer workflow for the DIA data analysis.

Recovery is only reported for the pesticides (in Figure 2). These are a simple target because they are massively concentrated and have characteristic MS2 spectra. What about the library for the AMD study? The reported number of 811 features seems very limited for this type of sample. There must be a bottleneck somewhere. A more thorough analysis is necessary to assess the capacity of the workflow to build assays for a normal sample. How many MS1 features were found? How many features map to a DB entry? For how many was it possible to obtain an informative MS/MS? For how many was it possible to generate an assay? Please elaborate.

We detected 4,460 features on average in the AMD DDA files with the *FeatureFinderMetabo*. 811 features were quantified at a 5% FDR. The reduction in feature number is associated with the filtering steps along the pipeline since the aim is to build a high-quality assay library. These filtering steps can also be disabled to some extent. Due to restrictions within SIRIUS, features with less than four fragment peaks in the MS2 spectra are filtered out. In addition, if SIRIUS' explained intensity score is below 85%, the feature is filtered out - since SIRIUS could not perform the fragment annotation correctly.

Further filtering depends on the parameterization of the workflow. Using solely known compounds, features without identification are filtered out before the linking process. While linking, features that do not show the same identification for a given threshold are filtered out. For example, suppose a feature was linked in 67 samples but was only correctly identified in 13 samples. In that case, the feature is filtered since it is not available with valid identification in, i.e., 50% of the samples. This step is performed to ensure a high-quality assay library based on the identification and fragment annotation. Unfortunately, since this is highly automated, we cannot give respective numbers for the individual filtering steps.

If we would build the library, allowing all features, with one mass trace with and without identification, and use the fragment annotation filter by SIRIUS we would end up with 7,520 possible features. In our work, the goal is not to achieve the highest possible number of features, but to achieve the highest possible accuracy in identification and quantification. As we apply relatively strict criteria for MS1 and MS2 quality, it is expected that the total number of high quality features is lower than without such strict filtering. However, we believe that these filters substantially increase the quality of the resulting data, which will be used for subsequent studies. Using the feature linking as described above on the feature number is reduced to 1,048 features in the library.

I don't know what to make of the simulation done with pesticides and NIST 17 (lines 135-144). As the authors state, libraries are experiment specific (line 64). The proposed simulation seems very far away from a real sample (i.e. the AMD study). Is it of any relevance to prove general quality of libraries? How can one verify the quality of the library in a more realistic setting, i.e. considering also dark matter, retention time, features that are not in the library, etc.?

As for Supp Figure 3B, it is not clear why 3 transitions should be preferred. The results for 25ppm/25ppm are almost identical between UIS1 and UIS3.

Thank you for this comment. The proposed simulation, as published individually ¹¹, uses realistic values for both the precursor m/z window and the fragment m/z window, selecting single query pesticide molecules to be tested against a combined complex background. For our experiment, we chose one of the most complex backgrounds available, the NIST 17 library. Although we are limited by known compounds present in the NIST library (which may or may not represent the complexity of the AMD dataset and dark matter), additional analyses were performed in our previous study (Figure 4 in ¹¹) using extrapolation to estimate how a more complex sample matrix would behave. These simulations show that while overall interferences increase, the relative performance of the different acquisition methods does not change. In addition to m/z , we expect orthogonal sources of information, such as retention time, to maximize unique detection. Unfortunately, due to lack of retention time information in commonly used spectral libraries such as NIST, we could not model the rate of co-elution in our simulations and we believe that predicting RT would currently be too inaccurate to produce reliable results. In our previous study, we modeled the effects of reducing “the analysis space” to a smaller retention time window (see Figure 4 in ¹¹) where we decrease the size of the background matrix thus removing compounds (as one would when focusing on a smaller retention time area in the chromatogram).

In Supplementary Figure 3B, we assessed the theoretical saturation of unique compounds for various methods. In practical applications, sample matrices will vary and assuming that a sample matrix consists of all NIST 17 compounds may either be too optimistic or pessimistic for a particular application. In this particular case, using pesticides, we agree that one or two transitions could have been used, but conservatively we chose three transitions for this measure in preparation for future applications with sample matrices that may have increased complexity. This is additionally supported in our previous study, where we demonstrate that DIA methods (with accurate MS1 information) are close to saturation already when using the three most abundant transitions (see Figure 3A in ¹¹).

Supplementary Material:

Based on these overall results and our previous study¹¹, scoring is based on both MS1 and MS2 information, in addition to retention time and fragment ion relative intensity .

Minor points:

- Text structure: the text lacks a clear organization. Adding Sections headings would be a start.

We added section headers to improve the readability and the documents organization.

- The limitations of DIAMetAnalyzer should be moved from the supplement to the main text.

We moved the limitations of the Pipeline into a subcategory to the main text.

Main text:

Limitations and runtime of the DIAMetAlyzer

It can be deemed as a limitation that DDA and DIA data has to be measured for an experiment. The main purpose of the DIAMetAlyzer workflow is to perform accurate quantification in a targeted manner. Here, the DDA data - for example - reference standards would be measured once to construct the assay library. This library can then be reused for DIA data analysis measured with the same experimental setup. In a targeted setting, it is generally necessary to invest resources to build accurate assays in order to achieve high-quality targeted results. While DDA is generally biased towards high abundant analytes, this will not impact measurements of low complexity, such as pure standards. When building assay libraries from complex samples, the library will be biased towards highly abundant analytes. We suggest to counteract this bias by enhancing such assay libraries with reference compounds measured from pure standards.

DIAMetAlyzer uses SIRIUS for fragment annotation, so the limitations in terms of high-resolution instruments and molecular masses of SIRIUS apply to the workflow as well. High mass compounds can in some cases not be processed by SIRIUS in a timely manner. The user can set a threshold of 100 s (default) per compound, to restrict the runtime. As a reference, the assay library generated from 67 DDA samples, with prior MS1 identification took around 2.5 h using 10 cores (Intel(R) Xeon(R) Gold 6140 CPU @ 2.30GHz). With allowing unknown features it took around 12.5 h using 28 cores. The runtime of the complete KNIME workflow for the targeted pesticide mix experiment, using one core (Intel Core i7 @ 3.50 GHz), was 36 minutes. All runtime improvements of SIRIUS in the future will also impact the runtime of the workflow.

- Line 18: “meaningful knowledge” > “meaningful data”

We think that meaningful biological knowledge in the abstract is the more suitable term, since we would like to analyze the “meaningful data” to gain “meaningful knowledge”.

- Line 111: “only high quality assays were included in the library” > how is quality defined?

We define the high-quality assay using DDA MS2 spectra for transitions extraction, in combination with our filtering steps, the SIRIUS annotation/filtering, feature linking and - in our case - by using three transitions.

- Line 193-195: the sentence is incomplete.

We corrected the incomplete sentence.

- Line 428: what is the "peak group level"?

OpenSWATH² performs transition extraction in a user-parameterised retention time window (e.g., 100s) (Supplementary Fig. 20). Multiple possible peak group candidates can be available for one analyte entry in the assay library. These peak group candidates are scored and ranked. We processed the highest scoring peak group candidate (peak group level/rank = 1) in our analysis.

Main text:

In the following, features with an FDR of 0.05 and highest scoring peak group (rank 1) were used for post-processing analysis.

- LIPID MAPS > LIPIDMAPS

We renamed the instances - thank you!

Reviewer #2:

Sorry for the long wait - I was swamped. The editor did a wonderful job following up with me and really tried to get a timely review out of me :). One of the main reasons that untargeted metabolomics analysis cannot be truly used at scale is because, unlike proteomics or sequencing or other omics technologies, there are no good ways to control FDR. I had reviewed this paper for NMeth and thought it was a good paper then and think that this is a good paper now. I -and I am sure the authors even more- was disappointed in the decision at NMeth as one of the main reasons for rejection was that a reviewer picked one inaccuracy of an annotation (which could have been textually addressed) and extrapolated that the method is incorrect. That was a complete misunderstanding by the reviewer of FDR and the role FDR plays in metabolomics analysis. The whole point of FDR estimation is so one can judge how many inaccuracies they are willing to accept before including them in the downstream analysis and biological interpretation. By definition of FDR, there will be inaccuracies. Proteomics, sequencing and many other OMICS fields utilize and rely on such FDR estimators. I am convinced that this paper will spur the development of other FDR methods and over time. Thus the field will collectively, and as a community, learn how to best do this in metabolomics. In short, this is an important paper, it's really the only method for the type of metabolomics data (DIA) that currently exists and thus is a key milestone in my opinion. It is also a key milestone as it will be foundational to push the ability to perform DIA metabolomics at the population scales - in which data independent analysis will play a key role in the future as it is more reproducible than DDA, the only other data format for which FDR exists. The impact of this paper will bear out long-term not short term as 99% of metabolomics labs don't know how to use or think about FDR and this requires continuous education of the entire community. Perhaps the authors should get together with many other top labs and write a review/perspective on the importance of FDR in metabolomics. The work is nicely done and thanks for providing such a great resource to the community, especially now it's not only available as KNIME functionality!

Thank you!

References

1. Guo, J. & Huan, T. Comparison of Full-Scan, Data-Dependent, and Data-Independent Acquisition Modes in Liquid Chromatography-Mass Spectrometry Based Untargeted Metabolomics. *Anal. Chem.* **92**, 8072–8080 (2020).
2. Röst, H. L. *et al.* OpenSWATH enables automated, targeted analysis of data-independent acquisition MS data. *Nat. Biotechnol.* **32**, 219–223 (2014).
3. Röst, H. L. *et al.* OpenMS: a flexible open-source software platform for mass spectrometry data analysis. *Nat. Methods* **13**, 741–748 (2016).
4. Benjamini, Y. & Hochberg, Y. Controlling the False Discovery Rate: A Practical and Powerful Approach to Multiple Testing. *Journal of the Royal Statistical Society: Series B (Methodological)* vol. 57 289–300 (1995).
5. Elias, J. E. & Gygi, S. P. Target-decoy search strategy for increased confidence in large-scale protein identifications by mass spectrometry. *Nat. Methods* **4**, 207–214 (2007).
6. Scheubert, K. *et al.* Significance estimation for large scale metabolomics annotations by spectral matching. *Nat. Commun.* **8**, 1494 (2017).
7. Wang, X. *et al.* Target-Decoy-Based False Discovery Rate Estimation for Large-Scale Metabolite Identification. *J. Proteome Res.* **17**, 2328–2334 (2018).
8. Reiter, L. *et al.* mProphet: automated data processing and statistical validation for large-scale SRM experiments. *Nat. Methods* **8**, 430–435 (2011).
9. Teleman, J. *et al.* DIANA—algorithmic improvements for analysis of data-independent acquisition MS data. *Bioinformatics* **31**, 555–562 (2015).
10. Rosenberger, G. *et al.* Statistical control of peptide and protein error rates in large-scale targeted data-independent acquisition analyses. *Nat. Methods* **14**, 921–927 (2017).

11. Shanthamoorthy, P., Young, A. & Röst, H. Analyzing Assay Specificity in Metabolomics Using Unique Ion Signature Simulations. *Anal. Chem.* **93**, 11415–11423 (2021).

Reviewers' Comments:

Reviewer #1:

Remarks to the Author:

The authors have addressed all critical concerns. The revised version suffers less from the long history of this manuscript for the benefit of readability.

It is a pity that the authors didn't take the opportunity to compare DIA with DDA. It would have been a landmark publication in the field of metabolomics. Nonetheless, DIAMetalyzer constitutes an important advance in the field and the community will greatly appreciate this contribution.

Response to reviewers' comments

Reviewer #1 (Remarks to the Author):

The authors have addressed all critical concerns. The revised version suffers less from the long history of this manuscript for the benefit of readability.

It is a pity that the authors didn't take the opportunity to compare DIA with DDA. It would have been a landmark publication in the field of metabolomics. Nonetheless, DIAMetalyzer constitutes an important advance in the field and the community will greatly appreciate this contribution.

Thank you very much for your assessment. We are happy that you regard DIAMetalyzer as an important advance in the field. We agree that a comprehensive comparison of DIA and DDA in the future would be very beneficial for the direction of the metabolomics field.